# Allograft Vesicoureteral Reflux after Kidney Transplantation

**DOI:** 10.3390/medicina58010081

**Published:** 2022-01-05

**Authors:** Alessandra Brescacin, Samuele Iesari, Sonia Guzzo, Carlo Maria Alfieri, Ruggero Darisi, Marta Perego, Carmelo Puliatti, Mariano Ferraresso, Evaldo Favi

**Affiliations:** 1General Surgery and Kidney Transplantation, Fondazione IRCCS Ca’ Granda Ospedale Maggiore Policlinico, 20122 Milan, Italy; brescacin.alessandra@gmail.com (A.B.); samuele.iesari@gmail.com (S.I.); soniaguzzonew@gmail.com (S.G.); ruggerodarisi@gmail.com (R.D.); martaperego9@gmail.com (M.P.); evaldofavi@gmail.com (E.F.); 2Pôle de Chirurgie Expérimentale et Transplantation, Institut de Recherche Expérimentale et Clinique, Université catholique de Louvain, 1200 Brussels, Belgium; 3Nephrology, Dialysis and Transplantation, Fondazione IRCCS Ca’ Granda Ospedale Maggiore Policlinico, 20122 Milan, Italy; carlo.alfieri1@gmail.com; 4Department of Clinical Sciences and Community Health, Università degli Studi di Milano, 20122 Milan, Italy; 5Division of General Surgery, Transplant Surgery Unit, Parma University Hospital, 43126 Parma, Italy; puliattic@gmail.com

**Keywords:** kidney transplant, vesicoureteral reflux, urinary tract infection, outcomes, allograft survival, systematic review

## Abstract

Allograft vesicoureteral reflux (VUR) is a leading urological complication of kidney transplantation. Despite the relatively high incidence, there is a lack of consensus regarding VUR risk factors, impact on renal function, and management. Dialysis vintage and atrophic bladder have been recognized as the most relevant recipient-related determinants of post-transplant VUR, whilst possible relationships with sex, age, and ureteral implantation technique remain debated. Clinical manifestations vary from an asymptomatic condition to persistent or recurrent urinary tract infections (UTIs). Voiding cystourethrography is widely accepted as the gold standard diagnostic modality, and the reflux is generally graded following the International Reflux Study Committee Scale. Long-term transplant outcomes of recipients with asymptomatic grade I-III VUR are yet to be clarified. On the contrary, available data suggest that symptomatic grade IV-V VUR may lead to progressive allograft dysfunction and premature transplant loss. Therapeutic options include watchful waiting, prolonged antibiotic suppression, sub-mucosal endoscopic injection of dextranomer/hyaluronic acid copolymer at the site of the ureteral anastomosis, and surgery. Indication for specific treatments depends on recipient’s characteristics (age, frailty, compliance with antibiotics), renal function (serum creatinine concentration < 2.5 vs. ≥ 2.5 mg/dL), severity of UTIs, and VUR grading (grade I-III vs. IV-V). Current evidence supporting surgical referral over more conservative strategies is weak. Therefore, a tailored approach should be preferred. Properly designed studies, with adequate sample size and follow-up, are warranted to clarify those unresolved issues.

## 1. Introduction

Vesicoureteral reflux (VUR) represents one of the most frequently observed urological complications of kidney transplantation (KT) [1,2,3]. It is defined as an abnormal flow of urine backward from the bladder to the ureter or, in extreme cases, up to the renal pelvis. In the general population, VUR can be either congenital (primary) or acquired (secondary), following blockage or failure of the bladder musculature, as well as dysfunction of the nerves controlling bladder emptying [4]. Clinical features range from an asymptomatic condition to end-stage renal disease (ESRD) and appear as directly related to the severity of the reflux and the occurrence of urinary tract infections (UTIs) [5,6,7]. The technique adopted for ureteral implantation and bladder function, at the time of transplant, certainly are key factors in the development of post-operative VUR. The correlation between post-transplant VUR and increased risk of UTIs is undisputed. Nonetheless, the transplant community has yet to reach an agreement on optimal pre-emptive strategy, clinical relevance, management, and impact on long-term allograft function and survival [8].

## 2. Literature Research

We performed systematic research to identify epidemiology, clinical presentation, diagnosis, staging, prognosis, treatment options, and follow-up strategies for VUR in adult KT recipients. We used the electronic databases PubMed and Embase. No time limits were applied, but we included studies published before 30 September 2021. We chose the following MeSH terms for the research through PubMed: “kidney”, “transplantation”, and “vesicoureteral reflux”. We used the following string to search through Embase: (‘transplantation’/exp AND ‘kidney’/exp AND ‘vesicoureteral reflux’/exp). Inclusion criteria were: all kinds of articles, including conference papers, but excluding case reports, all languages. Exclusion criteria were: species (non-human), age (paediatric recipients), and type of article (case reports). This topic has rarely been met with standard screening and reporting criteria, which prevented us from performing meta-analyses. As a consequence, we also excluded 251 articles because they were not amenable to a meaningful qualitative analysis. The selection process is displayed in Figure 1.

## 3. Epidemiology, Etiology, and Risk Factors for Post-Transplant VUR

The incidence of VUR after KT extends from 0.5 to 86%. This remarkably wide range depends on the highly variable criteria that are employed to screen KT recipients for VUR and on the reporting methodology [9,10].

Over the years, several factors have been recognized as increasing the risk of post-transplant VUR. They can be sorted in two main categories: modifiable and unmodifiable risk factors. Unmodifiable risk factors include recipient-related characteristics, such as sex, urinary tract abnormalities, or neurological disorders, whereas modifiable risk factors are surgical technique and surgeon’s expertise (Table 1) [11].

The association between sex and urological complications has been the subject of multiple investigations. Most literature suggests that females may be more prone to allograft VUR and UTIs [11,12,13,14]. However, in a study specifically designed to assess the effect of sex on VUR, Farr et al. observed that, on multivariate analysis, no sex-specific differences could be detected [15].

Further independent risk factors for post-transplant VUR are recipient age, non-Caucasian ethnicity, hypertension, type 2 diabetes mellitus, lower urinary tract abnormalities, and continuous or intermittent bladder catheterization [14]. Among the others, bladder conformation is probably the most important determinant. Indeed, recipients who have been on renal replacement therapy (RRT) for years, very often exhibit a hypo-compliant bladder with low capacity, thin walls, and high intra-vesical pressure [16]. Such a condition may represent a technical challenge when performing the ureteral implantation and may increase the risk of peri-operative complications [17]. In a recent study on 408 KT recipients, investigating possible relationships between duration of RRT, bladder capacity, and post-transplant urological complications, it has been observed that the incidence of VUR is significantly higher in dialysis vintage patients with an atrophic bladder (capacity < 50 mL) than those who have a bladder capacity of 50 mL or above, thus raising the hypothesis that atrophic bladder could actually predispose to VUR [18]. Inoue et al. have reported conflicting results [19,20]. In a study published in 2011, the authors found that intra-vesical pressure influenced the prevalence of VUR, which was higher among recipients with small atrophic bladders [19]. Nevertheless, in 2016, the same group showed that there were no statistically significant differences in VUR or other post-transplant urological complication rates between patients with or without a hypo-compliant bladder [20]. Such discrepancy has been arbitrarily ascribed to modifications in the surgical technique. In particular, it has been postulated that the difference in VUR rates was related to the adoption of a 3-mm sub-mucosal ureteral tunnel during the second study, which minimized the occurrence of VUR [20].

Multiple options for ureteral implantation are currently available. When the Lich–Grégoir technique is chosen, a single cystostomy is performed and the anastomosis is carried out between the distal ureter and the bladder’s mucosa; the detrusor muscle is oversewn to provide an anti-reflux tunnel [21,22]. The Politano–Leadbetter technique requires two cystostomies. The transplant ureter is introduced into the bladder through the first cystostomy, tunnelled for several millimetres, and eventually anastomosed to the bladder’s mucosa through the second cystostomy; the detrusor muscle is approximated to create an anti-reflux mechanism [23]. In the full-thickness technique, the ureter is anastomosed to the bladder’s wall, with full-thickness stitches, without tunnelling [24]. In case the U-stitch technique is preferred, one or two stitches are thrown at the distal tip of the ureter, brought through the bladder’s wall, and tied [25]. If the recipient has no ureteral abnormalities, an uretero-ureteral anastomosis between the allograft ureter and the native ureter can be also considered, ensuring a natural anti-reflux mechanism [14]. The Lich–Grégoir and the Politano-Leadbetter are the most frequently adopted techniques. The vast majority of transplant surgeons seem to favour the use of a relatively short ureter and the construction of a large uretero-cystostomy over a tunnelled reimplant, in an effort to reduce the likelihood of ureteral strictures [26]. However, while it is clear and intuitive that operative technique is the main cause of VUR, no general consensus exists on which type of ureteral implantation should be performed during KT. As pointed out by Duty et al., it is plausible that the occurrence and severity of the reflux may be influenced by the particular approach chosen for the ureteral anastomosis [26]. A systematic review and meta-analysis has compared the Lich–Grégoir vs. the Politano–Leadbetter and the Lich–Grégoir vs. the U-stich techniques. Remarkably, the Lich–Grégoir uretero-vesical anastomosis has resulted in fewer post-operative urological complications, including VUR [27]. Regardless of the reconstruction technique, it has become common to insert self-retaining vesicoureteral stents during the procedure. While this practice has proved successful in reducing urinary leaks, indwelling stents can increase the rate of UTIs [9]. What is less clear is whether a risk of VUR, although unlikely, remains after their removal. 

The possible impact of surgical expertise on the development of post-transplant VUR has also been evaluated with mixed results. Cash et al. have investigated the effect of surgeons’ experience on overall surgical complication rates after KT. In their study, no differences were observed between experienced and inexperienced surgeons [28]. On the contrary, a multivariate analysis by Farr et al. has shown that surgical experience is the most significant predictor of post-transplant VUR, even after multiple adjustments for potential confounders and sex-specific urological complications [15].

## 4. Clinical Presentation, Diagnosis, and Grading of Post-Transplant VUR

The main complication of VUR, after KT, is the development of UTIs. Accordingly, the most frequently observed clinical manifestations are dysuria, strangury, urinary frequency, urinary urgency, and fever. Post-transplant UTIs are classified into one of the following categories: asymptomatic bacteriuria, lower UTI, acute graft pyelonephritis (AGPN), and urosepsis. Asymptomatic bacteriuria is defined as the isolation of a bacterial strain with no symptoms of lower or upper UTI, including leukocyturia. Lower UTI involves bacteriuria associated with urinary symptoms (dysuria, urinary frequency, or urinary urgency) and/or mild fever (<38 °C), in the absence of the criteria for AGPN. AGPN requires significant bacteriuria and high fever (≥38 °C) and may be associated with allograft tenderness and acute transplant dysfunction. The diagnosis of urosepsis is made when the same bacterial strain can be isolated in simultaneous blood and urine cultures. UTIs can be also classified as new infections, relapses, or re-infections. Relapse is defined as the isolation of the same microorganism, which caused the first infection, in urine cultures obtained two or more weeks after completion of antibiotic treatment; re-infections are UTIs caused by a new agent that is different from the one isolated during the course of the previous infection [29].

In the vast majority of patients, the presence of allograft VUR can be easily detected before clinical onset. Even though the necessity and timing of VUR screening after KT remain debated, Hotta et al. have demonstrated that a voiding cystourethrography (VCUG), performed in the very early post-transplant phase (namely, at the time of bladder catheter removal), is a safe and feasible option [18]. Current policy is to wait for persistent or recurrent UTIs before proceeding with a diagnostic work-up, which may include a pelvic Doppler ultrasound (US) scan, a contrast-enhanced abdomen computed tomography (CT) scan, or a VCUG.

The gold standard imaging modality for both diagnosis and grading of VUR after KT is certainly a VCUG. This fluoroscopic study is obtained by introducing an iodinate contrast medium into the bladder via a temporary catheter, and it can easily detect VUR, as well as other bladder or urethral abnormalities. An US scan is often performed for prompt evaluation of the allograft. In asymptomatic recipients without hydronephrosis or impaired bladder emptying, no further investigation is generally carried out. In case of symptomatic UTIs with allograft dysfunction, a contrast-enhanced CT scan or a VCUG may be required [18,30]. According to the International Reflux Study Committee Scale, post-transplant VUR can be graded as: Grade I, reflux of urine limited to the ureter; Grade II, reflux of urine into the ureter, renal pelvis, and calyces without dilation; Grade III, reflux of urine that causes a mild-to-moderate dilation of the ureter, renal pelvis, and calyces with minimal blunting of the fornices; Grade IV, moderate ureteral tortuosity and dilation of renal pelvis and calyces; Grade V, gross dilation of the ureter, renal pelvis, and calyces, with severe swelling and ureteral twisting (Table 2) [31].

## 5. Management of Post-Transplant VUR

When and how to treat a post-transplant VUR have yet to be clarified (Table 2). 

The clinical management of VUR occurring in a kidney allograft depends on the grade of the reflux and, most importantly, on the severity and frequency of UTIs. Briefly, the following options are available: wait-and-see strategy [12], antibiotics administration [26], endoscopic injection of dextranomer/hyaluronic acid copolymer into the ureteral implantation site [32], pyelo-ureterostomy between the pelvis of the allograft and the native ureter [33], or ureteral re-implantation [34]. In case of asymptomatic VUR, the most reasonable option is to leave the reflux untreated. For symptomatic low-to-moderate VUR, endoscopic copolymer injection can be considered, in institutions with adequate expertise. The procedure was first described in 1995, and it is routinely performed under local anaesthesia or sedation. The copolymer is cystoscopically injected into the sub-mucosal layer of the bladder, at the site of the ureteral anastomosis, to expand the tunnel surrounding the ureter and, therefore, reduce the reflux. In the original study, the reflux was resolved in 58% of patients after the first injection, while this rate increased to 79% after two injections [32]. These encouraging results have been confirmed by other studies, reporting primary treatment success rates as high as 63% in both living- and deceased-donor KT [35]. In patients with symptomatic moderate-to-high grade VUR, clinical management is based on a variety of factors. Whang et al. recommend a tailored approach, considering allograft function, frequency of recurrent UTIs, compliance with antibiotic treatment, age, and overall health status of the recipient [12]. According to these authors, the degree of the reflux should not be regarded as the major determinant of the need for reconstructive surgery. In patients with a serum creatinine concentration ≥ 2.5 mg/dL, antibiotic suppression without planned end date should be preferred because allograft function is already compromised. Similarly, for elderly or frail recipients, as well as those refusing surgery, prolonged antibiotic administration is advocated. In all other cases, patients should be referred for surgical evaluation. The choice of the surgical procedure primarily depends on the availability of a non-refluxing ipsilateral ureter [36]. Whenever possible, an uretero-ureterostomy between the allograft ureter and the native ureter represents the best option. In fact, using the native ureter provides a better anti-reflux mechanism and easier access, should endoscopic evaluation or treatment be necessary in the future [33]. The anastomosis can be performed in either an end-to-end or an end-to-side fashion, depending on the length and size of the transplant ureter. Urological stents are usually inserted in both ureters before the operation to reduce the risk of unintentional damage during dissection. When a suitable native ureter is not available, a new uretero-cystostomy is performed using the transplant ureteral stump, possibly adopting the Politano–Leadbetter technique [12]. It is also worth mentioning the technique proposed by Turunça et al., which consists of an extra-vesical seromuscular tunnel lengthening [37]. This procedure aims to increase the length of the seromuscular tunnel to more than 3 cm to cover the distal segment of the transplanted ureter. Comparison between tunnel lengthening, uretero-ureterostomy, and pyelo-ureteral anastomosis have shown lower surgical complication and recurrent UTIs rates, better allograft function, shorter operative times and hospital stays following tunnel lengthening [37]. Overall, excellent results have been described after reconstructive surgery, regardless of the specific technique chosen for ureteral reimplantation. Patil et al. have reported that 95% of the patients who had undergone surgery were relieved from symptomatic UTIs and freed from antibiotic suppression [36]. However, considering the risk associated with general anaesthesia and surgery, some authors suggest a trial of prophylactic antibiotics in all patients with mild-to-moderate symptoms and Grade I-III reflux [26]. While the role of prophylaxis against *Pneumocystis jirovecii* pneumonia with trimethoprim-sulfamethoxazole is not an option for the treatment of VUR itself, it has been shown that prophylactic antibiotics reduce the frequency of bacterial infections in KT recipients, including UTIs. Consequently, the last KDIGO guidelines recommended prophylactic trimethoprim–sulfamethoxazole for six-twelve months after transplant [8]. Since UTIs are a significant source of morbidity in recipients with VUR, it can be argued that indefinite prophylaxis with trimethoprim-sulfamethoxazole might be of some use in this category, though the available evidence is poor. 

As a last suggestion, biofeedback therapy might constitute a promising approach to VUR, at least for recipients with proven dysfunctional voiding. The role of biofeedback, intended as the set of practices meant to establish appropriate bladder control and micturition, has not been explored in KT, and this lack of evidence makes it worth studying biofeedback treatments in prospective trials.

## 6. Impact of Post-Transplant VUR on Allograft Function and Survival

The impact of VUR on transplant-related outcomes is still unclear [10,38,39]. Margreiter et al. performed one of the largest studies assessing the association between VUR and allograft survival in both living- and deceased-donor KT. They used a standardized per-protocol VCUG to evaluate 646 consecutive recipients before hospital discharge. The prevalence of VUR was 41%. One year after transplant, the estimated glomerular filtration rate was significantly lower in patients with VUR than controls. However, such a difference was not confirmed at three and five years of follow-up. Sorting patients according to the severity of the reflux did not affect results, as recipient and allograft survival, as well as cell-mediated and antibody-mediated rejection rates remained comparable [30]. In a similarly designed study with a smaller sample size, our group also demonstrated that VUR did not influence long-term allograft function and survival. However, this conclusion is limited to Grade I-III VUR, as in the population included there were no patients with Grade IV-V VUR [40]. Coulthard et al. have used the dimercaptosuccinic acid (DMSA) scan to evaluate allograft morphology, function, and scarring in recipients with or without VUR. In up to 40% of the patients with documented VUR and UTIs, a specific pattern of photon-deficient areas could be observed. Such peculiar scarring was associated with irreversible loss of function in half of the subjects [41]. Obha et al. reported long-term allograft dysfunction in KT recipients with VUR and AGPN [42]. The detailed information, arising from the systematic review of the literature, is fully reported in Table 3 and Table 4. 

## 7. Conclusions

VUR represents a leading urological complication of KT. Atrophic bladder is the most relevant predisposing factor, and it is predominantly related to pre-existing urological disorders or dialysis vintage. The surgical technique used for ureteral implantation is considered as another possible determinant, but supporting evidence is scarce. Post-transplant VUR should be graded according to the International Reflux Study Committee Scale. The application of a rigorous and consistent grading system is recommended for proper data collection and analysis. Clinical presentation is extremely variable. The vast majority of recipients with VUR is asymptomatic. Main complaints are related to the development of persistent or recurrent UTIs. To date, there are no standardized screening protocols. Nevertheless, VCUG is widely accepted as the gold standard diagnostic modality. Clinical guidelines for the management of VUR after KT are lacking. Reasonably, asymptomatic patients can be followed-up with a wait-and-see strategy. Symptomatic subjects, with complex comorbidities or irreversible allograft dysfunction, as well as those refusing more invasive options, are preferably treated with prolonged antibiotic suppression. For symptomatic recipients, with preserved transplant function and low-to-moderate VUR, endoscopic copolymer injection may be considered. In case of endoscopic treatment failure or severe VUR, patients should be offered surgical reconstruction.

Over the last two decades, a number of studies have been performed to evaluate incidence, risk factors, and consequences of VUR after KT. Several diagnostic protocols and treatment strategies have also been proposed. Despite this great effort, evidence remains weak in any relevant aspect of the topic. In particular, it is of paramount importance to clarify whether VUR may jeopardize long-term allograft function and survival. Future research will have to minimize the multiple bias affecting previous studies. Timely identification of recipients with VUR, at higher risk of UTIs, will likely improve outcomes, as recurrent UTIs represent the leading complication of VUR and a well-recognized risk factor for post-transplant morbidity and mortality.

## Figures and Tables

**Figure 1 medicina-58-00081-f001:**
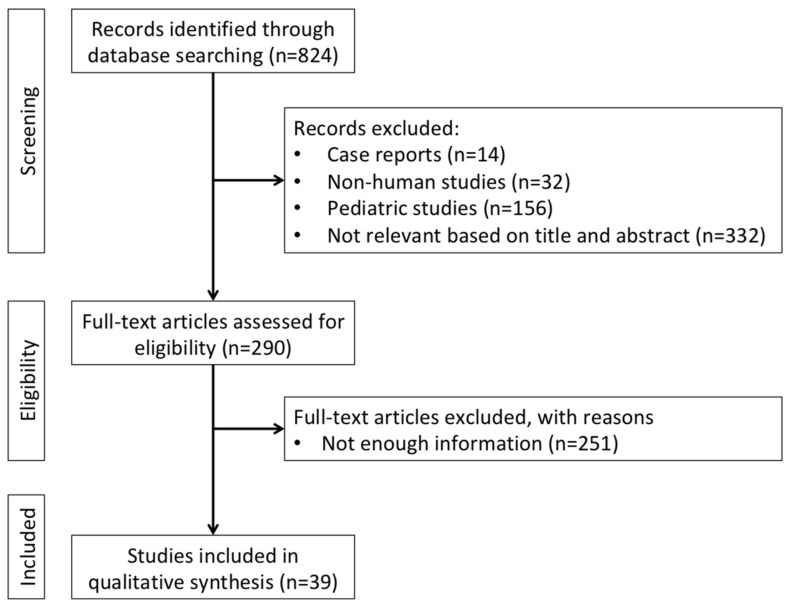
Flowchart of literature research and study selection.

**Table 1 medicina-58-00081-t001:** Risk factors for post-transplant vesicoureteral reflux.

Unmodifiable Risk Factors
Female sex
Non-Caucasian ethnicity
Age at transplant
Lower urinary tract abnormalities
Atrophic bladder
Dialysis vintage
Hypertension
Type 2 Diabetes mellitus
**Modifiable risk factors**
Surgical technique for ureteral implantation
Surgical expertise

**Table 2 medicina-58-00081-t002:** Diagnosis and management of post-transplant vesicoureteral reflux.

Diagnosis	Pros	Cons
Doppler US scan	ready-to-use	non-functional evaluation
	non-invasive	operator dependent
Contrast-enhanced CT scan	high resolution	non-functional evaluation
	reproducible	contrast-induced nephropathy
		radiation exposure
Voiding VCUG	gold standard	availability/expertise
		lack of standardized protocols
		radiation exposure
**Management**	**Pros**	**Cons**
Wait and see	non-invasive	risk of future infections
Antibiotic suppression	non-invasive	does not affect reflux
		antibiotic resistance
		drug-induced side effects
Endoscopic polymer injection	minimally invasive	Grade I-III reflux
	high success rate	availability/expertise
	repeatable	
Ureteral reimplantation	very high success rate	invasive
	Grade IV-V reflux	expertise

Abbreviations: US, ultrasound; CT, computed tomography; VCUG, voiding cystourethrography.

**Table 3 medicina-58-00081-t003:** Epidemiology of kidney transplant recipients with vesicoureteral reflux.

Authors	Year	Period	Participants (n)	Type of Population	Cases with VUR (n, M/F)	Incidence (%)	Mean Age at KT (Years)	Mean HD Duration (Months)	Cases with Symptoms (n)
Mathew TH et al. [43]	1975	-	72	recipients	27	38	-	-	-
Lucas BA et al. [38]	1979	1972–1975	112	allografts	-	<10	-	-	-
Matrosimone S et al. [10]	1993	1985–1991	103	recipients	89 (63/26)	86	40	65 ± 10	4
Ostrowski M et al. [9]	1999	1984–1996	39	recipients	12	31	-	-	0
Kmetec A et al. [44]	2001	-	23	recipients with UTI	16	70	39	-	23
Ohba K et al. [42]	2004	1990–2001	131	renal biopsies	7	5	-	-	12
Praz V e al. [45]	2005	1979–1999	277	allografts	4	1	45	-	-
Coulthard MG et al. [41]	2006	1994–2005	30	recipients	19	63	10	-	-
Jung GO et al. [39]	2008	2005–2006	75	recipients	47	63	42	36 vs. 30	-
Nie ZL et al. [2]	2009	1993–2007	1223	recipients	14	1	-	-	-
Favi E et al. [40]	2009	-	37	recipients	15 (9/6)	41	41 ± 13	-	-
Kayler L et al. [24]	2010	review	-	-	-	-	-	-	-
Whang M et al. [1]	2011	1993–2009	2548	recipients	78 (16/72)	3	-	-	78
Gołębiewska J et al. [7]	2011	2009–2010	89	recipients	7	8	48 ± 14	25 ± 24	58
Inoue T et al. [19]	2011	2010–2011	101	recipients	30	30	42	32	-
Obara T et al. [46]	2012	1998–2006	164	recipients	36	22	46	60	-
Sandhu K et al. [47]	2012	2000–2009	-	-	2	-	31	-	-
Dinckan A et al. [14]	2013	2000–2008	1673	recipients	60 (28/32)	4	-	-	-
Margreiter M et al. [30]	2013	1999–2007	646	recipients	263	41	53	-	-
Marzi VL et al. [48]	2013	2002–2012	14	recipients	2	14	38	13	-
Gołębiewska J et al. [13]	2014	2007–2009	209	recipients	-	-	46 ± 14	-	-
Farr A et al. [15]	2014	2001–2007	598	recipients	237 (167/70)	40	54	-	237
Alberts VP et al. [27]	2014	review	-	-	-	-	-	-	-
Gołębiewska J et al. [29]	2014	2007–2009	209	recipients	19	9	48 ± 14	29 ± 35	19
Riediger C et al. [49]	2014	2001–2009	646	allografts	10 (3/7)	2	55	-	-
Duty BD et al. [26]	2015	review	-	-	-	-	-	-	-
Di Carlo HN et al. [34]	2015	review	-	-	-	-	-	-	-
Inoue T et al. [20]	2016	2009–2012	61	recipients	16	26	-	22	-
Choi YS et al. [35]	2016	2000–2014	853	recipients	24	3	-	-	-
Soliman M et al. [50]	2016	2013–2014	203	allografts	1	0	-	-	-
Hotta K et al. [18]	2017	1996–2011	347	recipients	191	55	43 ± 14	-	32
Turunç V et al. [37]	2017	2010–2014	812	recipients	38 (26/12)	5	45	-	38
Nane I et al. [51]	2017	1983–2017	789	allografts	9	1	-	-	9
Sui W et al. [11]	2018	2005–2013	9038	recipients	99	1	52 ± 14	-	-
Yang KK et al. [33]	2019	2011–2018	262	recipients	-	-	-	-	3
Gutiérrez-Jiménez AA et al. [52]	2019	2010–2018	23	recipients with UTI	23 (10/13)	100	34	-	23
Whang M et al. [12]	2020	1993–2016	3890	recipients	168 (44/124)	4	48	-	-
Di Lascio G et al. [53]	2020	2017–2019	84	allografts	84	100	-	-	-
Ladhari N et al. [54]	2021	2007–2018	209	allografts	31 (19/12)	15	28	12	10

Abbreviations: VUR, vesicoureteral reflux, M, male; F, female; KT, kidney transplantation; HD, haemodialysis; UTI, urinary tract infection.

**Table 4 medicina-58-00081-t004:** Clinical features of kidney transplant recipients with vesicoureteral reflux.

Authors	Clinical Manifestation	Diagnosis	Classification	Grade (n)	Treatment (n)	Success (%)	Graft Survival ^a^ (%)	Loss of Function	Patient Survival ^a^ (%)
Mathew TH et al. [43]	-	-	-	-	-	-	-	-	-
Lucas BA et al. [38]	-	-	-	-	-	-	58	-	-
Matrosimone S et al. [10]	recurrent UTIs hypertension	VCUG	Ransley PG et al. [54]	I-II (62) III (27)	antibiotics	-	100	no	99
Ostrowski M et al. [9]	-	VCUG	-		-	-	-	-	-
Kmetec A et al. [44]	recurrent UTIs	VCUG	-	I-II (9) III (7)	-	-	-	-	-
Ohba K et al. [42]	pyuria pyelonephritis	VCUG	-	-	-	-	-	-	-
Praz V et al. [45]	-	-	-	-	-	-	-	-	-
Coulthard MG et al. [41]	UTIs nephropathy	VCUG MAG3 DMSA scan	-	-	-	-	-	-	-
Jung GO et al. [39]	UTIs rejection loss of function	VCUG	IRCS	I (6) II (22) III (17) IV (2)	-	-	-	--	
Nie ZL et al. [2]	-	VCUG	-	-	antibiotics (8) surgery (6)	93	100		88
Favi E et al. [40]	UTIs, loss of renal function	VCUG	IRCS	I II III	-	-	91	no	84
Kayler L et al. [24]	-	-	-	-	-	-	-	-	-
Whang M et al. [1]	UTIs	VCUG	-	-	Antibiotics copolymer injection surgery	26, 0, 74	-	-	-
Gołębiewska J et al. [7]	lower UTIs upper UTIs bacteremia	-	-	-	antibiotics	-	-	-	-
Inoue T et al. [19]	loss of function	VCUG	IRSC	-	-	-	-	-	-
Obara T et al. [46]	-	VCUG	-	-	-	-	-	-	-
Sandhu K et al. [47]	AGPN	-	-	-	ureteropyelostomy (native ureter)	100	100	no	100
Dinckan A et al. [14]	recurrent UTIs	VCUG	-	III (8) IV (40) V (12)	reconstruction	-	100	-	-
Margreiter M et al. [30]	UTIs loss of function	VCUG	IRCS	I (51) II (128) III (66) IV (18)	-	-	-	yes	-
Marzi VL et al. [48]	-	VCUG	-	-	-	-	-	-	-
Gołębiewska J et al. [13]	-	-	-	-	-	-	-	-	-
Farr A et al. [15]	UTIs hydronephrosis	VCUG US	-	I (46) II (120) III (54) IV (17)	-	-	-	-	-
Alberts VP et al. [27]	-	-	-	-	-	-	-	-	-
Gołębiewska J et al. [29]	recurrent UTIs AGPN sepsis	-	-	-	-	-	-	-	-
Riediger C et al. [49]	-	-	-	-	ureteropyelostomy (native ureter)	100	80	no	90
Duty BD et al. [26]	-	-	-	-	-	-	-	-	-
Di Carlo HN et al. [34]	-	-	-	-	-	-	-	-	-
Inoue T et al. [20]	-	VCUG max IV pressure	IRSC	I-II (11) III-IV (5)	-	-	-	-	-
Choi YS et al. [35]	UTIs	VCUG	-	-	copolymer injection	71	-	-	-
Soliman M et al. [50]	-	-	-	-	medical treatment	-	-	no	-
Hotta K et al. [18]	UTIs graft scarring	VCUG	-	-	-	-	-	-	-
Turunç V et al. [37]	recurrent UTIs	VCUG	IRCS	-	surgery	95	100	-	100
Nane I et al. [51]	-	-	-	-	-	-	-	-	-
Sui W et al. [11]	UTIs	-	-	-	-	-	-	-	-
Yang KK et al. [33]	AGPN	VCUG	IRCS	III (3)	copolymer injection surgery	0 100	-	-	-
Gutiérrez-Jiménez AA et al. [52]	AGPN	VCUG	IRCS	II (3) III (11) IV (9)	copolymer injection	78	74	-	-
Whang M et al. [12]	UTIs loss of function	VCUG	-	-	antibiotics copolymer injection surgery	- 0 -	-	-	-
Di Lascio G et al. [53]	-	VCUG CEUS	-	I (18) II (47) III (7)	-	-	-	-	-
Ladhari N et al. [54]	recurrent UTIs	-	-	-	-	-	60	-	-

^a^ by the end of the study period. Abbreviations: AGPN, acute graft pyelonephritis; DMSA, ^99m^ Tc-dimercaptosuccinic acid; IRSC, International Reflux Study Committee; MAG3, ^99m^ Tc-mercaptoacetyltriglycine; CEUS, (contrast-enhanced) ultrasonography; UTI, urinary tract infection; VCUG, voiding cystourethrography; VUR, vesicoureteral reflux; IV intra-vesical.

## Data Availability

Not applicable.

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
