# Peer review of "Allograft Vesicoureteral Reflux after Kidney Transplantation"

_medicina, 2022, doi:10.3390/medicina58010081_

Round 1

Reviewer 1 Report

Comments to the Authors

Brescacin et al. conducted an excellent and comprehensive review organizing data on vesicoureteral reflux after kidney transplantation. In detail, this review summarizes information on epidemiology, risk factors, clinical presentation, and management as well as the impact on allograft function and survival. Although vesicoureteral reflux displays a major complication following kidney allograft transplantation, a consensus on risk factors and clinical management is missing in the field. The authors present in detail the state of research as well as gaps of knowledge for the direction of future research in the field.

Major comments:

  • In the introduction, the authors mention the etiology of vesicoureteral reflux in non-transplant population but miss out to elaborate the etiology of vesicoureteral reflux in patients with kidney transplant allografts in detail. Please give more background information on vesicoureteral reflux after kidney transplant.

Minor comments:

  • The authors do not mention current guidelines for screening and treatment of urinary tract infections post kidney transplant which can be linked to vesicoureteral reflux. Please briefly mention a state-of-the-art guideline, e.g., KDIGO 2009 Transplant Recipient Guideline.

Author Response

Dear Editors,

We are grateful to the Reviewers for their constructive comments.

Also, we are pleased to have the opportunity to submit to Medicina a revised version of the manuscript entitled “Allograft vesicoureteral reflux after kidney transplantation”.

As requested, we have revised the paper according to the comments and suggestions provided.

All changes have been explained in a point-by-point response and tracked in the text. In the interest of clarity, comments (Q) are reported in italics and answers (A) follow in regular font.

Reviewer #1

Q1) Brescacin et al. conducted an excellent and comprehensive review organizing data on vesicoureteral reflux after kidney transplantation. In detail, this review summarizes information on epidemiology, risk factors, clinical presentation, and management as well as the impact on allograft function and survival. Although vesicoureteral reflux displays a major complication following kidney allograft transplantation, a consensus on risk factors and clinical management is missing in the field. The authors present in detail the state of research as well as gaps of knowledge for the direction of future research in the field.

A1) Thanks for this encouraging comment.

Q2) Major comments: In the introduction, the authors mention the etiology of vesicoureteral reflux in non-transplant population but miss out to elaborate the etiology of vesicoureteral reflux in patients with kidney transplant allografts in detail. Please give more background information on vesicoureteral reflux after kidney transplant.

A2) In the original version of the manuscript, we have reserved an entire section to the root-cause analysis of VUR after kidney transplantation: 3. Epidemiology and risk factors for post-transplant VUR. However, as requested, we have modified the introduction section and the section 3 in order to make it clearer.

Q3) Minor comments: The authors do not mention current guidelines for screening and treatment of urinary tract infections post kidney transplant which can be linked to vesicoureteral reflux. Please briefly mention a state-of-the-art guideline, e.g., KDIGO 2009 Transplant Recipient Guideline.

A3) As suggested, the aforementioned guideline has been added.

As a last comment, we made a mistake when entering in your submission system the name of the third co-author, whose right name is “Sonia” and the right surname is “Guzzo”. Can you please remove “Maria”?

We did our best to address all the comments and improve the manuscript.

We truly hope that our article can now be accepted for publication in Medicina.

We look forward to hearing from you.

Kindest regards,

Evaldo Favi MD, PhD and Mariano Ferraresso, MD, PhD, Full Professor

Kidney Transplantation, Fondazione IRCCS Ca’ Granda Ospedale Maggiore Policlinico

Department of Clinical Sciences and Community Health, University of Milan

Milan, Italy

Reviewer 2 Report

Dear Dr. Brescacin,

in your wortk, you summarize important aspects of VUR after KTx, e.g. independent risk factors or impact of specific surgical procedures of ureteral implantation.

Nevertheless some additional clarifications should be addressed to enhance relevance and impact of the paper:

1) can the authors assume the impact of a ureteral splint, which is quite commonly used after kidney transplantation especially at earlier time points, for the incidence and severity of VUR ;

2) in the early phase after KTx transplantation centers commonly use Trimethoprim/Sulfamethoxazol for PcP prophylaxis; can the authors argue on the “add-on” impact of Trimethoprim/Sulfamethoxazol for UTI in VUR patients after KTx

3) regarding the management of VUR can the authors argue on the impact of biofeedback therapy for transplanted pts., e.g. micturate according to predefined time intervals, e.g. every 2 hours

Author Response

Dear Editors,

We are grateful to the Reviewers for their constructive comments.

Also, we are pleased to have the opportunity to submit to Medicina a revised version of the manuscript entitled “Allograft vesicoureteral reflux after kidney transplantation”.

As requested, we have revised the paper according to the comments and suggestions provided. All changes have been explained in a point-by-point response and tracked in the text. In the interest of clarity, comments (Q) are reported in italics and answers (A) follow in regular font.

Reviewer #2

Q1) Dear Dr. Brescacin, in your work, you summarize important aspects of VUR after KTx, e.g. independent risk factors or impact of specific surgical procedures of ureteral implantation. Nevertheless some additional clarifications should be addressed to enhance relevance and impact of the paper: can the authors assume the impact of a ureteral splint, which is quite commonly used after kidney transplantation especially at earlier time points, for the incidence and severity of VUR;

A1) Ureterovesical double-J stents were introduced to cut ureteral complications after kidney transplantation and their use served the purpose. The presence of an indwelling stent causes reflux by definition, especially at higher bladder fill volumes. However, the “stent” effect is supposed to last the time of stent retention, while stents are typically removed four to six weeks after kidney transplant. Concordantly, in the literature we reviewed the presence of VUR is screened after stent removal. At any rate, we commented in the text that the role of double-J stents in preventing early urological complications is acquired whereas their role in causing later VUR in less clear.

Q2) in the early phase after KTx transplantation centers commonly use Trimethoprim/Sulfamethoxazol for PcP prophylaxis; can the authors argue on the “add-on” impact of Trimethoprim/Sulfamethoxazol for UTI in VUR patients after KTx

A2) While the role of PCP prophylaxis with trimethoprim-sulfamethoxazole is not an option for the treatment of VUR itself, it has been indeed shown that prophylactic antibiotics reduce the frequency of bacterial infections in kidney transplant recipients, including urinary tract infections. Based upon these evidences, the last KDIGO guideline for the care of kidney transplant recipients recommended prophylactic trimethoprim–sulfamethoxazole for six-twelve months after kidney transplant. Since urinary tract infections are a significant source of morbidity in recipients with VUR, it can be argued that indefinite prophylaxis with trimethoprim-sulfamethoxazole might be of some use in this category, though the available evidence is poor. Still, we have included these considerations in the text.

Q3) regarding the management of VUR can the authors argue on the impact of biofeedback therapy for transplanted pts., e.g. micturate according to predefined time intervals, e.g. every 2 hours

A3) Thanks for this stimulating suggestion. No evidence is indeed available in the literature about the role of biofeedback therapy in approaching vesicoureteral reflux and its consequences in kidney transplant recipients. While this makes it hard to extensively discuss this strategy in the context of kidney transplantation, we have introduced it as a promising pathway to explore for future research.

As a last comment, we made a mistake when entering in your submission system the name of the third co-author, whose right name is “Sonia” and the right surname is “Guzzo”. Can you please remove “Maria”?

We did our best to address all the comments and improve the manuscript.

We look forward to hearing from you.

Kindest regards,

Evaldo Favi MD, PhD and Mariano Ferraresso, MD, PhD, Full Professor

Kidney Transplantation, Fondazione IRCCS Ca’ Granda Ospedale Maggiore Policlinico

Department of Clinical Sciences and Community Health, University of Milan

Milan, Italy

Reviewer 3 Report

Please consider expanding the methods section to elaborate on the reasons for excluded studies. 

Line 66: may consider changing to modifiable and unmodifiable variables. 

Author Response

Dear Editors,

We are grateful to the Reviewers for their constructive comments.

Also, we are pleased to have the opportunity to submit to Medicina a revised version of the manuscript entitled “Allograft vesicoureteral reflux after kidney transplantation”.

As requested, we have revised the paper according to the comments and suggestions provided. All changes have been explained in a point-by-point response and tracked in the text. In the interest of clarity, comments (Q) are reported in italics and answers (A) follow in regular font.

Reviewer #3

Q1) Please consider expanding the methods section to elaborate on the reasons for excluded studies.

A1) As requested, we have detailed the reason for the exclusion of a number of studies after thorough full-text appraisal.

Q2) Line 66: may consider changing to modifiable and unmodifiable variables.

A2) Done (text and table).

As a last comment, we made a mistake when entering in your submission system the name of the third co-author, whose right name is “Sonia” and the right surname is “Guzzo”. Can you please remove “Maria”?

We did our best to address all the comments and improve the manuscript.

We truly hope that our article can now be accepted for publication in Medicina.

We look forward to hearing from you.

Kindest regards,

Evaldo Favi MD, PhD and Mariano Ferraresso, MD, PhD, Full Professor

Kidney Transplantation, Fondazione IRCCS Ca’ Granda Ospedale Maggiore Policlinico

Department of Clinical Sciences and Community Health, University of Milan

Milan, Italy

This manuscript is a resubmission of an earlier submission. The following is a list of the peer review reports and author responses from that submission.

Round 1

Reviewer 1 Report

Brescacin et al presented review entitled: ‘’Allograft vesicoureteral reflux after kidney transplantation’’.

General comment:

I read this report with interest but unfortunately I am very disappointed with poor design and quality. This report is in range with an average student’s book. The standard is to perform review of literature and if possible meta-analysis. That would be of great interest for the readers, and this narrative review is the lowest level of significance and scientific journal with high impact should not even consider report like this for publication. The main question is: ‘’What benefits readers benefit from this report?!’’.

Specific comments:

  • There is no visible strategy of literature search
  • How many databases were searched, when?
  • Which number of articles were identified, which were excluded, which number of articles are included?
  • What was the main outcome?
  • Which parameters were studied?
  • The authors should identify some important parameters and extract them from literature and perform an analysis. In this form this is of extremely minor value.

Considering all above mentioned I cannot support publication of this report. The only way to improve this review is to perform review of literature (learn how to do it) and possible meta-analysis. Narrative reposts may be suitable only for non-academic Journals or students / residents books.

Reviewer 2 Report

The authors performed review of literature regarding allograft vesicoureteral reflux after kidney transplantation

Unfortunately, I do not see personal input of the author in this report, or it is very week, most of the paragraphs in this report are repetition of well-known facts from the literature.

This topic has been published many times and many similar reviews exist in medical literature. On the other hand there is also systematic review published on the same topic.

This review is more like a chapter from a book and not a real review of literature. The authors should review crucial clinical factors/symptoms and perform review on each (compare findings from large series of patients).

The conclusions drawn from the authors are general and well known and have been reported many times.

I do not see any benefit for the readers from this review. Most of this can be found books and published articles.